# Analysis of Digital Literacy in Health through Active University Teaching

**DOI:** 10.3390/ijerph18126674

**Published:** 2021-06-21

**Authors:** Alejandro De la Hoz, Javier Cubero, Lina Melo, Miguel A. Durán-Vinagre, Susana Sánchez

**Affiliations:** 1Health Education Lab, Department of Experimental Sciences and Mathematics Education, University of Extremadura, 06011 Badajoz, Spain; aldelahoz@alumnos.unex.es (A.D.l.H.); lvmelo@unex.es (L.M.); 2Department of Psychology, University of Extremadura, 06011 Badajoz, Spain; mduranv@unex.es (M.A.D.-V.); ssanchez@unex.es (S.S.)

**Keywords:** eHealth literacy, confidence, higher education, health promotion

## Abstract

As we have seen as a consequence of the COVID-19 global pandemic, our scientific-technological society requires a transformation of knowledge in a global, digital and virtual manner. In light of this, and to improve the public health skills of professionals working to promote health education, one of the current priorities is to train pre-service teachers on how to search for health-related scientific knowledge relating to bio-health through digital literacy in health. The objectives of this study were to determine the level of eHealth literacy, scientific knowledge in health and confidence on the part of students, analyzing the degree of improvement following a teaching intervention with students of the Bachelor’s Degree of Education of the University of Extremadura. A quasi-experimental longitudinal study was carried out with pre-test and post-test groups and a mixed data analysis. It involved the application of an active cooperative methodology with tutoring using the jigsaw technique, with the use of four quality digital browsers to enhance scientific rigor. The results show that the initial level of eHealth literacy and confidence was problematic. Following the intervention with the cooperative methodology, both these levels and the level of scientific knowledge in health reached an excellent level. In conclusion, this suggests that university training programs need to be implemented to improve digital and scientific skills in health education as they are currently insufficient. It has been demonstrated that a cooperative active methodology improved these levels and accordingly its use in active and digital higher education should be promoted.

## 1. Introduction

Students in a global and digital society require scientific training, particularly in health-related areas. The generation of problematic scientific knowledge due to poor digital literacy in health (eHealth literacy) is a daily occurrence which affects our quality of life. This fact has become even more apparent in recent times in relation to the efforts to counter the COVID-19 pandemic declared by WHO, caused by the virus now known as Severe Acute Respiratory Syndrome Coronavirus 2 (SARS-CoV-2) [1,2].

One of the priority lines of research in science teaching is biomedical science literacy. Nowadays, scientific knowledge is generated, modified and developed rapidly due to digitalization and use of the internet [3,4].

The vast majority of health-related internet searches are made using terminology with concerning deficiencies [5,6,7], leading to misconceptions due to lack of scientific and digital skills [7], which can cause or aggravate individual and community health problems.

In addition, the professional skills of teacher trainees in the field of public health, particularly health promotion, are obtained through health education (EpS) courses [8,9]. This is an area which requires constant training and updating of health knowledge. Therefore, students need training in the management of digital tools and resources to guarantee the independence and rigor of self-learning processes and enhance their skills in an up-to-date and effective manner.

Health literacy (HL) in the context of health education (HE) is a concept that determines a person’s capacity and motivation to access, understand and use health-related information. HL is linked to healthy lifestyle habits due to decision-making about the individual’s health needs. Accordingly, a person with a low literacy level is more likely to have health-related problems [10].

In the school environment, these concepts assume new meanings and responsibilities [11]. However, there is no specific key competency in the field of health, and it is diluted across the curriculum, raising the need for specific proposals as to what and how to organize and develop health issues in schools [12].

As a result of this need recognized by entities such as WHO, UNESCO, the Council of Europe, the European Commission, the Spanish Ministry of Health, Consumer Affairs and Social Welfare and the Spanish Ministry of Education, the COMSAL Project was developed to define these competencies [13]. In addition, scientific societies and international organizations (ASE, ESERA, OECD) point to health and a healthy lifestyle as one of the basic foundations of education. In 2015, the United Nations approved the document “Transforming Our World: The 2030 Agenda for Sustainable Development” [14], comprising 17 Sustainable Development Goals (SDGs) to be achieved by 2030.

In this scenario, the acquisition of health skills is fundamental at all educational stages as a means of improving people’s quality of life and ability to act in accordance with current scientific and technological knowledge. Therefore, science teaching should also promote training in health literacy [12,15].

Consequently, the term digital literacy in health arises, which is defined as the ability to search, find, understand and evaluate health information from electronic sources and apply the knowledge acquired to address or solve a health problem [16]. Digital literacy in health modifies the health beliefs of individuals and also contributes to improve their quality of life by enhancing their ability to access information relevant to health care for both themselves and their families [17].

A recent study [18] found that only 12% of the 74 research papers analyzed conducted interventions to improve low literacy levels. Education is necessary to ensure future teachers have the adequate digital literacy in health and knowledge of quality resources that improve their skills for searches and analysis of health issues on the internet [19].

On the other hand, learning is influenced by the affective domain of the students. This perspective has not been the subject of recent consideration in the field of education or teacher training. Students can influence their learning through metacognition. It has been demonstrated that there is a loss of motivation and negative emotions when they feel less competent with this scientific content [20].

Metacognitive control must be self-regulated through self-confidence, which refers to the level of confidence when carrying out a task in a global or general way, [21] and not in a specific way such as self-efficacy [22]. The Pew Report shows that 9 out of 10 people believe the information they find on the internet. This poses a risk to human health, due to a lack of training in the skills needed to carry out an adequate critical analysis of digital information and the resources where it is found [6].

Training should also be supported by active strategies such as cooperative learning [23,24,25]. Future teachers need to learn new and diverse methodologies where the students are the fundamental axis of the teaching–learning process. This cooperative methodology has many advantages, including significant improvements to learning, attitudes and values, increased motivation and greater responsibility and interdependence [26].

Consequently, it is necessary to provide trainee teachers with quality tools so that they can help their future students to have a good level of digital literacy in health. The advantages gained through use of ICT and virtual education [27] and strengthening group work [28] should also be incorporated, with priority being placed on learning methodologies where students are the fundamental axis of the teaching-learning process.

In light of all the above, the purpose of our project is to determine the level of digital literacy in health, scientific knowledge in health and confidence with its use, analyzing the improvement in these levels following a cooperative intervention for both male and female trainee teachers of the Bachelor’s Degree of Education of the University of Extremadura.

## 2. Materials and Methods

### 2.1. Study Design

The study had a quasi-experimental longitudinal design with pre-test and post-test groups and a mixed data analysis: a quantitative approach with descriptive and inferential statistics, and, on the other hand, a qualitative approach with management of qualitative categories of digital literacy in health.

### 2.2. Participants

The study population was selected based on convenience and non-probabilistic sampling. It consisted of university students studying bioscientific training subjects under the Bachelor’s Degree in Primary Education of the Faculty of Education of the University of Extremadura during the 2016/2017 (*n* = 28) and 2017/2018 (*n* = 14) academic years. A total of 42 students participated—28 female and 14 male—at all times according to the recommendations and bioethical principles of the Helsinki Declaration and subsequent international agreements [29].

### 2.3. Variables and Instrument

In order to determine the level of digital literacy in health, the Spanish version of the eHEALS questionnaire [30] was completed before and after the intervention. The eighth item of this questionnaire was also used to determine the level of confidence of the students.

The level of scientific knowledge in health was analyzed using the specific question-naire on the Zika virus already previously used in other studies [31], with four open questions regarding: specific bioscientific knowledge, prevention, health promotion and digital resources. This questionnaire was issued at the end of the intervention and was revised and validated by Health Education teachers. It was only used at the end (post-test) of the intervention because, following a previous generic oral screening, it was observed that the participants had no previous scientific knowledge regarding the Zika virus.

### 2.4. Intervention

The same active intervention was incorporated in the practical seminar classes of the bioscientific training subjects, during three hours in both academic years. These interventions carried out by the same teacher were developed applying an active cooperative methodology (tutoring) using Aronson’s jigsaw technique [25]. Firstly, they were instructed on the knowledge and use of certain resources such as search engines and online bibliographic databases that stand out due to their quality or scientific prestige. These digital resources were both in Spanish (Dialnet^®^, Medline Plus^®^, Familia y Salud^®^) and in English (Healthfinder^®^) and all of them have been recommended by the literature in this field [19,32,33].

The students were divided into groups of four members, with four modalities/functions distributed between the members. Thus, each person became an expert of one of the resources; after a period of time, the experts of each resource met up to agree and ‘learn by doing’ through the exchange of information. Finally, the group met as they did at the beginning of the activity so that each member could share their experience and learning with the others. The teacher provided guidance and collaborated when the groups needed additional information or in the case of incomplete groups.

### 2.5. Data Analysis

With respect to the data analysis, Microsoft Office Excel 365 software (Redmond, WA, USA) was used for the descriptive analysis of the data gathered in the questionnaires used in this study and to determine the averages (X), frequencies (F), percentages (%) and standard deviations (SD) of each item. Meanwhile, for the inferential analysis the SPSS (Statistical Package for the Social Sciences) 25.0 statistical software (IBM, Madrid, Spain) was used. After concluding from the Kolmogorov–Smirnov test that the data did not follow a normal distribution, inferential statistical tests were applied: the Mann–Whitney test to check for significant differences (*p* < 0.05) between the sexes of the students, and the Wilcoxon test to check for the existence of significant differences (*p* < 0.05) between the pre-test and post-test groups.

The qualitative analysis of digital literacy in health was carried out based on four qualitative categories: Inadequate, Problematic, Sufficient and Excellent, from the recent study of a university in Mexico with a population similar to that of our study [34]. These four qualitative categories were used to analyze the results of the eHealth Literacy Scale (eHEALS) questionnaire [30], which allowed us to determine the level of eHealth literacy of the sample (Table 1).

To evaluate the level of scientific knowledge in health, the four questions of the Zika virus questionnaire were analyzed according to the standards correct, semi-correct and incorrect, with scores of 1 point, 0.5 points and 0 points, respectively, in order to determine the average result from 0 to 4 for the four open questions. Those averages were then converted into percentages.

Four qualitative categories were established to determine the level of scientific knowledge in health: Inadequate, Problematic, Sufficient and Excellent. These four qualitative categories were applied in accordance with the results obtained from our own scale based on the percentage of successful responses, allowing us to determine the level of scientific knowledge in health (Table 2).

## 3. Results

### 3.1. Analysis of Digital Literacy in Health

Table 3 shows that the average pre-test value of digital literacy in health for the samples analyzed was 23.93 ± 4.59, with female students having a higher average than male students. The post-test results confirm this trend, with an average value for the questionnaire of 34.55 ± 2.69.

The results of the Mann–Whitney U test indicate that there are no statistically significant differences (*p* < 0.05) between the female and male pre-service teachers, neither in the pre-test—with a level of significance of 0.58—nor in the post-test—with 0.06—although this latter case was borderline. Wilcoxon’s statistical test indicates that there are statistically significant differences (*p* < 0.05) between the pre-test and post-test results, both globally and by sex (0.00).

More specifically, the eighth item of the questionnaire, “I am confident with the use of internet information to make health decisions”, which allowed us to analyze the level of confidence, follows the same trend with no statistically significant differences between sexes and with statistically significant differences between the pre-test—2.71 ± 0.86—and the post-test—4.21 ± 0.78.

According to the four qualitative categories of eHealth literacy, the pre-test level is Problematic (20–24), while the post-test level is Excellent (32–40).

### 3.2. Analysis of Scientific Knowledge in Health

Following analysis of the questionnaires with the scoring system mentioned above of correct (1 point), semi-correct (0.5 points) and incorrect (0 points), the average was 3.59 ± 0.32, with female students having a higher average than male students. The results of the Mann–Whitney U test indicate that there are statistically significant differences (*p* < 0.05), with a value of 0.01 (Table 4).

This average of 3.59 out of 4 represents an average success in the responses of 89.75%. According to the four qualitative categories used to determine the level of scientific knowledge in health (Table 2), the level of the sample after the intervention is Excellent.

Broken down by question, it can be seen in Table 5 that questions 2 and 4 were answered by the entire sample correctly, while questions 1 and 3 were not completed correctly.

Finally, the results of question 4 (“Indicate one rigorous and effective scientific publication or resource for your knowledge”) were analyzed in order to determine the most commonly used resources. Two digital resources stood out, Medline Plus^®^ and CDC (Center for Disease Control and Prevention), which were cited 11 times. After these, Dialnet^®^ and WHO appeared eight times. Finally, both Familia y Salud^®^ and Healthfinder^®^ were cited five times.

## 4. Discussion

The results of this study reveal the low level of professional skills that future teachers currently possess in the field of public health.

The initial average value for digital literacy in health was 23.93, placing it at the Problematic level (20–24), which is a similar result to other studies [7,31,34]. On the other hand, some studies [8] have shown higher average levels which are associated with a high level of education. This may be a reflection of the inadequate training provided in the Spanish education system as highlighted in international reports [35] and the lack of training in this area in university teacher education programs. In the case of the Bachelor’s Degree in Primary Education at the University of Extremadura where this research was carried out, the subject “Motor Activity, Physical Activity and Health” is the only one related to health promotion. This is also an optional course in one of the studies cited [36]. Thus, it is not seen as a priority and is dependent upon the teacher’s intention to focus on this area in the classroom, which explains the lack of level of knowledge on the part of the university students. Accordingly, specific training proposals should be made as to what and how to organize and develop health promotion in university teacher training degrees [9,10,12].

The results at the end of the intervention, with an average value of 34.93, clearly show that university training in eHealth through interventions based on a cooperative active methodology (tutoring) is an effective means of improving searches for information and use of digital resources relating to health science knowledge. This average equates to an Excellent level (32–40), an improvement on other previous studies where such methodologies were also applied [7].

Recent educational trends emphasizing skills-based training such as Learning to Learn and the Future of Education and Skills 2030 by the OECD [14] identify three skills that students must develop: cognitive and metacognitive skills; social and emotional skills; and physical and practical skills. The first of these areas includes the development of Learning to Learn skills, while the second includes self-confidence. In this sense, special mention should be made here of the evolution of the confidence levels, which improved significantly following the intervention as in the case of other similar studies [31].

In terms of the scientific knowledge in health, the intervention was also effective. The sample had an average value of 89.75% success in the questionnaire, similar to the results of other interventions, which demonstrated an improvement in health competencies [31] and scientific literacy [25]. After the intervention, the level of scientific knowledge in health was Excellent.

As for the gender differences, there were no significant differences in the level of eHealth literacy, confirming the results of previous studies [7,34]. However, there were statistically significant differences in the level of scientific knowledge, in line with the results of similar studies. This may be due to two causes. Firstly, the number of participants in the sample of our study means there is little dispersion between the data, and when this number is increased, there would be no such differences. A second possible cause is the increased interest in health issues and learning styles, although a more detailed study would be needed to confirm this. These results coincide with those of other studies [7], which reported gender differences following an intervention to improve eHealth skills, with female participants obtaining higher results.

The results also show that students have used the resources suggested by the teacher —Dialnet^®^, Medline Plus^®^, Healthfinder^®^ and Familia y Salud^®^—and also had the opportunity to find and use information from other quality resources such as WHO and CDC. As the level of eHealth literacy increases, the use of various digital sources of health information is more common [18].

Accordingly, a cooperative methodology to improve digital literacy in health can be a useful classroom resource for both schools and universities responsible for the training of future teachers, as has been highlighted in previous studies [31,37].

## 5. Conclusions

The effectiveness of an active cooperative methodology with tutoring as a tool to promote health has been confirmed. Therefore, its use should be promoted in university teaching programs, with teachers assuming an active role as agents to promote public health.

The need for this paradigm change has become increasingly urgent due to the exceptional global consequences of the COVID-19 pandemic. Health promotion programs need to offer a wide variety of quality digital resources to search for health-related scientific knowledge, as well as training in the critical quality analysis of information sources [38]. Accordingly, websites should be designed and created to provide quality resources and adequate information on quality digital sources that allow students to improve their learning in the field of eHealth.

This study highlights the need to implement health education plans with the aim of improving future teachers’ professional health skills. Students need to learn how to search for information and use digital bio-health scientific knowledge resources, given that they currently have an inadequate level of digital literacy in health, scientific knowledge and confidence in this area.

We advocate incorporation of courses by the Department of Experimental Sciences and Mathematics in the Bachelor’s Degree of Education to improve training programs and foster self-learning with greater independence and rigor, as a means of improving health promotion in an up-to-date and effective manner.

Finally, the limitations of this study include the size of the sample used and the possible biases generated. The authors propose interventions with other active methodologies and in other social/health-related university populations as future lines of research. The importance of active training for the non-university population should also not be forgotten [37], given that we must constantly be on the alert to deal with new pandemics arising due to infectious diseases.

## Figures and Tables

**Table 1 ijerph-18-06674-t001:** Qualitative categories of eHealth literacy.

Categories	Scale
*Inadequate*	0–19
*Problematic*	20–24
*Sufficient*	25–31
*Excellent*	32–40

**Table 2 ijerph-18-06674-t002:** Qualitative categories of the scientific knowledge in health with Zika virus.

Categories	Percentage
*Inadequate*	0–25%
*Problematic*	26–50%
*Sufficient*	51–75%
*Excellent*	76–100%

**Table 3 ijerph-18-06674-t003:** Average and standard deviations from the overall results of the eHealth Literacy (eHEALS) questionnaire.

	Total	Female	Male
Pre-test	23.93 ± 4.59 *	24.36 ± 4.92	22.85 ± 4.35
Post-test	34.55 ± 2.69 *	34.93 ± 2.99	33.43 ± 2.62

* Statistically significant differences (*p* < 0.05).

**Table 4 ijerph-18-06674-t004:** Average and standard deviations from the overall results of the Zika virus questionnaire.

Sex	Average	Standard Deviation	Percentage
Female	3.76 *	0.28	94%
Male	3.25 *	0.30	81.25%
Total	3.59	0.32	89.75%

* Statistically significant differences (*p* < 0.05).

**Table 5 ijerph-18-06674-t005:** Frequency and percentage of responses in the Zika virus questionnaire.

	Question 1	Question 2	Question 3	Question 4
*Correct*	29 (69.05%)	42 (100%)	27 (64.29%)	42 (100%)
*Semi-correct*	12 (28.57%)		15 (35.71%)	
*Incorrect*	1 (2.38%)			
Total	42 (100%)	42 (100%)	42 (100%)	42 (100%)

## Data Availability

For our study could put in contact with all authors.

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
