# Peer review of "Analysis of Digital Literacy in Health through Active University Teaching"

_ijerph, 2021, doi:10.3390/ijerph18126674_

Round 1
Reviewer 1 Report
The subject of the study is interesting and in editorial line with the journal.
The abstract is descriptive of the work.
It is recommended to review the way of citing within the text using the format of the journal.
Introduction: It is surprising that they framed their work on the need to acquire those skills by COVID when the fieldwork of the study was done many years earlier, so it is rather quaint that they envisioned in 2016 that they would need those skills to face demands arising in 2020.
Methods. It should be named Materials and Methods as it says in the journal instructions. The subsections are very strange and confusing, they should follow the usual ones: design, participants, variables and instruments, procedure, data analysis. More description of that part is needed. It is not understood which students have been selected, because the size obtained in two academic years seems to be from a specific group, from a specific course, and not from a complete degree. How were they selected, by whom, how many in each course? Similarly, the procedure needs to be developed much further. If the reviewer wanted to replicate the study, she should know how each step was performed. It should be explained how the students were selected, at what time the instruments were given to them, if to all at the same time, if in the different courses the procedure was done at the same time, who does the intervention, in the two years it was done by the same teachers, etc. A lot of information is missing to understand how it was done and to be able to verify that no important biases have been committed. Exactly what information did they look for? Each subject autonomously? Organize the information. In the first two paragraphs they talk about instruments, then intervention, data analysis, and in the last two paragraphs they talk about instruments again. They should give complete information on the instruments, including their psychometric properties.
Results: I do not believe that they have been carried out adequately. With so few subjects, as they indicate, and divided into sexes, they should not use means and standard deviations.
There are data that should be in the method section in instruments, e.g. Table 2 and 4.
They present some results differentiated by sex when this was not the objective of the study. They only present pre-post intervention results of the e-HEALS questionnaire, but what about the other objectives? There are other data that do not differentiate by sex. It is not understood why they have done the analyses in this way.
Discussion: as the results are not understood, it is difficult to assess the coherence of the discussion.
Author Response
Dear Editor and Referees, May 28th, 2021
The authors are writing to thank you for your valuable opportunity to publish our manuscript (ijerph-1205828) in your prestigious scientific Journal. In addition, the authors thank to three Referees who have assessed our article for acceptance.
Therefore, we will try to modify all your valuable recommendations or answer convincingly to the constructive comments of your three Referees in each of the Sections.
Initially, clarify that the presentation format of the manuscript and all the bibliographic citations have been revised according to the Journal, and the English language by an official translator.
Introduction:
In particular regarding by Reviewer 2, we must explain the correct connection of our research with the problem of misinformation or Infodemic generated by the COVID-19 Pandemic. It is known to all that low levels of Health Literacy through two variables: -Knowledge and Confidence- in the digital information found causing the deficient prevention of SARS-CoV-2.
Both variables of the Health Literacy: Scientific Knowledge in Health and Confidence together with the exact level of eHealth Literacy have been the three objectives of our studie.
For this reason, the text referring to this intimate relationship of Health Literacy and COVID-19 prevention has been modified.
With respect to the proposed Objective and following the indications of the Editor and Referees, it has been redrafted. Its approach has been improved, the term affective domain is eliminated by Confidence and the variable sex is added. All this manages to improve the understanding of our Results and therefore of our study.
Methods:
Following the constructive comments of Referees 1 & 2, the corresponding subsections have been reorganized also following the indications of the Journal. And its explanation has been improved to better understand how the research was carried out and avoid biases in its replication.
Regarding the Participants:
Its sample number is clarified by academic year. It should be noted that they were selected for convenience from all the students of the same Bioscientific training subject of the 3rd University Year during 2 consecutive academic courses, being therefore 2 different groups of students, one for each year 2016 and 2017.
Regarding the Questionnaires:
In particular, eHEALS is a questionnaire already used in other prestigious international investigations and adapted to a Spanish version:
Norman, C.D.; Skinner, H.A. eHEALS: The eHealth Literacy Scale. J Med Internet Res. 2006; 8: e27
On the other hand, the Questionnaire on Scientific Knowledge in Health for the Zika Virus has already been performance and applied by the authors in another research. Before this questionnaire was designed and validated by expert Spanish university professors and researchers in Health Education.
Cubero, J.; Sánchez, S.; Vallejo, J.R.;, Luengo, M.L.;, Calderón, M.;, Bermejo, M.L. Cooperative learning for university training in health literacy. Journal of the Medical Education Foundation. 2018; 21 (2): 97-100.
Clarify that this specific questionnaire was only used at the end (postest) of the intervention because previously with a generic and oral screening it was observed that the previ-ous science knowledge about Zika Virus in these participants was null.
Following the valuable recommendations of the Referee 2, the required aspects of the intervention design (samples, academic courses, duration, recourses, where, who and duration) are expanded and clarified, thus being able to replicate the study
Results:
Previously the objective approach has been improved by eliminating the term Affective Domain for Confidence and the variable Sexes is added. All this manages to improve the understanding of our Results and therefore of our study.
In particular, the Results and their Tables are reorganized for better clarity and understanding, as well as their associated wording. And this section has improved its interpretation because the Methods it has been thoroughly revised.
Discussion and Conclusion
For Discussion Section, the authors think that after what was expanded and explained the previous Sections, the Discussion is correct and convincing.
And Conclusions following the recommendations of the Editor and Referee 2, the possible limitations generated in our study are added.
Finally, the authors propose our future lines of research.
Sincerely yours,
Prof. Dr. Javier Cubero
UEx. Badajoz. Spain
Reviewer 2 Report
In my opinion, the article addresses a relevant topic and especially necessary considering the current health concerns we face globally. However, the manuscript presents some shortcomings that need to be addressed before an eventual publication.
Overall, the article needs to be clearer and more accurate in the presentation of the data and the arguments. In addition to the authors’ revision of the content, I think that the text needs proofreading in English to ensure correctness, accuracy, and clarity.
INTRODUCTION
- The argument should be more clearly organized, following a logical line of discourse, and showing the gap you aim to fill in. For instance, you could first argue the importance of health literacy, health education, and digital literacy in health; then explain how it is covered (or not) in schools and faculties of education and which are the shortcomings; finally, explain how your research contributes to the field.
- I suggest revising the research objective to express it more clearly: is the objective to assess the level of digital literacy in health, scientific knowledge, and affective domain in a group of Education students? Is it to assess the effectiveness of a particular intervention to raise these levels?
METHODS
Different aspects of the data collection and analysis need to be addressed.
Regarding the research design:
- The timing of the study is not specified. You say it was conducted during two academic years, does it mean that the same procedure was implemented with two groups of students, one each year, or all students participated for two years? Or were they involved for less than a year (e.g. a semester)? Which was the duration of the intervention? Which was the gap time between pre and post-intervention measures?
- What you mean by “scientific knowledge in health” and “affective domain” in this study should be explained.
Regarding the intervention
- Which were the criteria used to select the information sources used in the intervention?
- Why did the “puzzle” method used in the intervention? Has been successfully used previously in previous studies or interventions in the field?
- Which were the instructions given to the students during the intervention? Which was the task they had to complete?
Regarding the instruments:
- The questionnaire to evaluate the level of Digital Literacy in Health the eHealth Literacy was created for this study? If so, how was it created? Did it already exist? If so, was it validated for the sample population? Which variables did it analyse?
- Why the scientific knowledge in health was evaluated using the theme of the Zika virus? Was it the same questionnaire or a different one from the one used to evaluate the level of Digital Literacy in Health the eHealth Literacy? If it is different, did it already exist? Was it created for this study? How was it created? Which variables did it analyse?
- It is not clear how qualitative data were collected (there were open questions in the questionnaire?) and how were they analysed? If the qualitative approach consisted just of dividing the results into four categories of score ranges, it is still quantitative data, not qualitative.
RESULTS
The results should be more clearly presented, and sufficient information should be provided to facilitate its interpretation. Particularly, in section 3.2, regarding the analysis of scientific knowledge in health, there is no comparison between pre and post-intervention measures, therefore the impact of the intervention cannot be assessed. In addition, table 3 is difficult to interpret: it would be easier if the information of the maximum possible score (4) was included in the table, and it is not clear what the data in column “Percentage” refers to. It is neither clear how to interpret the column percentage in the light of table 4.
DISCUSSION AND CONCLUSIONS
The discussion and conclusions sections should be revised considering the previous comments.
You can also consider a discussion on the specific impact of the intervention methodology (“puzzle” method) and the information sources used: to what extent was the improvement observed due to each intervention component?
Finally, you can also discuss the sustainability and transferability of the improvements: could the increased competence on health literacy that, according to your paper, was achieved be used by the students outside the university context? Does the intervention have a lasting effect, or can it fade with time? These can also be questions to be addressed by further research.
Author Response

(The authors gave the same response as above.)

Reviewer 3 Report
The state of the art clarifies and contextualizes the relevance of the subject analysed: eHealth Literacy. The analysis is well connected with the objectives of the research, well documented from relevant scientific literature and with the Sustainable Development Goals.
The objectives are clearly stated from the beginning and connects with the methodology.
The methodology is complete and well justified by the type of research carried out.
Regarding the results: sowing the low level of professional skills in Public Health among future teachers, the research provides information to improve teacher training and health literacy among citizens.
Author Response

(The authors gave the same response as above.)

Round 2
Reviewer 1 Report
The reviewer appreciates the effort made by the authors to take on board the reviewers' suggestions in order to make the work more readable for the journal's readers.
Author Response
June 8th, 2021
Dear Editor and Referees of IJERPH,
The authors are once again grateful for your decision regarding our revised manuscript (ijerph-1205828).
Again all the authors have tried to modify and correct their valuable comments.
Clarify that the wording and understanding of the objectives has been reviewed.
The authors of the manuscript, most of them experts in qualitative analysis and their research, believe that the treatment of qualitative analysis through the management and design of categories allows its correct naming and use.
And finally following your comments, clarify that all English language in the manuscript has once again been reviewed by an official translator.
Yours sincerely.
Prof. Dr. Javier Cubero
Health Education Lab
UEx. Spain

Reviewer 2 Report
The authors have made an effort to improve their manuscript and some of the suggestions made in the first revision have been addressed.
However, some other issues have not been addressed yet.
In my first revision, I suggested restructuring the introduction section and make the argument and the writing clearer. However, I see no substantial changes in this section.
The research objective has been rewritten, but it is not clearly written. I suggest simplifying it or splitting it into two objectives, for instance:
1) to determine the level of Digital Literacy in Health, Scientific Knowledge in Health and Confidence (in what? in their ability to search information?) of students of the Degree of Education at the University of Extremadura.
2) to know whether these levels of Digital Literacy in Health, Scientific Knowledge in Health and Confidence can be improved with a tutored cooperative intervention (on...)
In the methods section, it is not yet clear which were the instructions given to the students during the intervention, which was the task they had to complete, or which information they had to search.
In addition, as mentioned in my previous revision, I suggest not considering the “qualitative analysis” as qualitative, as after grouping the data into four categories the data is still quantitative, and percentages (quantitative) are used to analyse these categories.
Finally, it seems that the English writing has not been revised. Although I am not a native English speaker I think it should be improved. I have included some specific comments in the document but I suggest the whole manuscript be revised.

Author Response

(The authors gave the same response as above.)
